# A Risk Stratification System in Myeloma Patients with Autologous Stem Cell Transplantation

**DOI:** 10.3390/cancers16061116

**Published:** 2024-03-11

**Authors:** Wancheng Guo, Christopher Strouse, David Mery, Eric R. Siegel, Manit N. Munshi, Timothy Cody Ashby, Yan Cheng, Fumou Sun, Visanu Wanchai, Zijun Zhang, Clyde Bailey, Daisy V. Alapat, Hongling Peng, Samer Al Hadidi, Sharmilan Thanendrarajan, Carolina Schinke, Maurizio Zangari, Frits van Rhee, Guido Tricot, John D. Shaughnessy, Fenghuang Zhan

**Affiliations:** 1Myeloma Center, Department of Internal Medicine, Winthrop P. Rockefeller Cancer Institute, University of Arkansas for Medical Sciences, 4301 W. Markham St. Slot# 508, Little Rock, AR 72205, USA; wguo3@uams.edu (W.G.); dmery@uams.edu (D.M.); mnmunshi@uams.edu (M.N.M.); ycheng@uams.edu (Y.C.); fsun@uams.edu (F.S.); vwanchai@uams.edu (V.W.); zzhang4@uams.edu (Z.Z.); baileyclyde@uams.edu (C.B.); salhadidi@uams.edu (S.A.H.); sthanendrarajan@uams.edu (S.T.); cdschinke@uams.edu (C.S.); mzangari@uams.edu (M.Z.); vanrheef@archildrens.org (F.v.R.); gjtricot@uams.edu (G.T.); 2Department of Haematology, Second Xiangya Hospital, Central South University, Changsha 410011, China; penghongling@csu.edu.cn; 3Department of Medicine, University of Iowa, Iowa City, IA 52242, USA; christopher-strouse@uiowa.edu; 4Department of Biostatistics, University of Arkansas for Medical Sciences, Little Rock, AR 72205, USA; siegelericr@uams.edu; 5Department of Biomedical Informatics, University of Arkansas for Medical Sciences, Little Rock, AR 72205, USA; tcashby@uams.edu; 6Department of Pathology Clinical, University of Arkansas for Medical Sciences, Little Rock, AR 72205, USA; dvalapat@uams.edu

**Keywords:** multiple myeloma, autologous stem cell transplantation, prognosis

## Abstract

**Simple Summary:**

Autologous stem cell transplantation (ASCT) is a longstanding myeloma treatment, but patient outcomes vary. In a retrospective study of 5259 patients with multiple myeloma (MM) at the University of Arkansas, we identified adverse prognostic factors, including delayed MM-diagnosis-to-ASCT duration, high serum ferritin, and low transferrin levels. These findings may enhance existing prognostic models. We also pinpointed poor prognosis markers, such as high serum calcium and low platelet counts, albeit in a smaller patient subset. Utilizing seven accessible high-risk variables, we devised a four-stage system, validated in both the training dataset and an independent cohort of 514 ASCT-treated MM patients from the University of Iowa. This staging system’s robust validation underscores its potential clinical utility, providing insights into cytogenetic risk factors. The ATM4S system presents a practical approach to refine prognostic assessments and guide personalized treatment strategies in ASCT-treated MM patients.

**Abstract:**

Autologous stem cell transplantation (ASCT) has been a mainstay in myeloma treatment for over three decades, but patient prognosis post-ASCT varies significantly. In a retrospective study of 5259 patients with multiple myeloma (MM) at the University of Arkansas for Medical Sciences undergoing ASCT with a median 57-month follow-up, we divided the dataset into training (70%) and validation (30%) subsets. Employing univariable and multivariable Cox analyses, we systematically assessed 29 clinical variables, identifying crucial adverse prognostic factors, such as extended duration between MM diagnosis and ASCT, elevated serum ferritin, and reduced transferrin levels. These factors could enhance existing prognostic models. Additionally, we pinpointed significant poor prognosis markers like high serum calcium and low platelet counts, though they are applicable to a smaller patient population. Utilizing seven easily accessible high-risk variables, we devised a four-stage system (ATM4S) with primary stage borders determined through K-adaptive partitioning. This staging system underwent validation in both the training dataset and an independent cohort of 514 ASCT-treated MM patients from the University of Iowa. We also explored cytogenetic risk factors within this staging system, emphasizing its potential clinical utility for refining prognostic assessments and guiding personalized treatment approaches.

## 1. Introduction

Over the past three decades, there have been significant advancements in the treatment of multiple myeloma (MM), including the development of immunomodulators [1], proteasome inhibitors [2], targeted therapies [3], new chemotherapy combinations [4], autologous stem cell transplantation (ASCT), and immunotherapies [5]. ASCT provided hope for the cure of MM through high-dose chemotherapy and bone marrow reconstitution. However, it carries the risk of bone marrow suppression and infection. Despite various evaluations of age, performance status, and co-morbidities for ASCT eligibility [6,7,8,9], the outcomes of MM patients still vary significantly. At present, a range of indicators are utilized to assess the appropriateness of autologous stem cell transplantation for patients with multiple myeloma [10]. These encompass disease stage and activity, treatment response, physical condition and functional status, age, and comorbidities [6]. Autologous transplantation is typically considered for patients with stable or partially controlled disease, favorable treatment response, and relatively good physical condition without significant comorbidities, especially for younger and relatively healthy individuals [11].

Currently, multiple clinical indicators are considered relevant to the prognosis of multiple myeloma. C-reactive protein [12] and (CRP) β-2-microglobulin [13] (B2M) have been identified as robust risk indicators. Additionally, aspartate aminotransferase (AST) and lactate dehydrogenase [14] (LDH) have been established as effective prognostic factors. The international staging system (ISS) utilizes readily available clinical parameters, employing B2M and albumin (ALB) for staging purposes. Furthermore, traditional cytogenetic techniques, encompassing metaphase karyotyping and TriFISH, provide valuable insights into the identification and analysis of chromosomal abnormalities that are pertinent to the prognosis of multiple myeloma. Several models have been developed to recognize myeloma tumor burden [15], progression-free survival, and overall survival [16,17,18], among which the ISS-series systems are the most well-known [19,20,21]. These models assist in providing an answer to the question of the expected progression-free survival and overall survival of MM patients following initial treatment. However, it remains challenging to predict the risk of MM patients receiving ASCT.

To develop a convenient tool for classifying ASCT MM patients, we conducted a follow-up study of 5259 patients who received ASCT after 1989, with a median progression-free survival of 38.8 months and a median overall survival of 56.1 months. We collected and evaluated 29 clinical variables before transplantation (most of them were at first diagnosis), ultimately identifying the seven variables (ISS stage, diagnosis–transplant period, ferritin, transferrin, LDH (lactate dehydrogenase), age, and gender) used to generate a four-stage system for MM risk stratification prior to ASCT.

## 2. Materials and Methods

### 2.1. Study Design, Participants, and Clinical Variables

Twenty nine clinical variables: age, gender, race, isotype, serum light chain type, urine light chain type, transferrin, ferritin, iron, bone marrow plasma cell percentage (choosing the higher value in biopsy and aspiration), albumin, B2M (beta-2 microglobulin), LDH, creatinine, CRP (C-reactive protein), Hb (hemoglobin), platelets, monocytes, lymphocytes, serum M protein level, urine M protein, calcium, BMI (body mass index), serum glucose, cholesterol, triglycerides, HDL (high-density lipoprotein), LDL (low-density lipoprotein), and myeloma diagnosis–transplant period were collected before ASCT. The inclusion criteria for patient selection are shown in Figure 1. We excluded the LDL variable because it had a missing rate of more than 50%. The primary event endpoint was progression-free survival, and overall survival was secondary. The definition of survival/progression and the calculation of PFS/OS time are provided in the Appendix A. The collection of all data was approved by the Institutional Review Board of the University of Arkansas for Medical Sciences, and written informed consent was obtained from all subjects for the procurement of samples following the guidelines outlined in the Declaration of Helsinki.

### 2.2. GEP Score Calculation and Chromosome Translocation Prediction by GEP

Bone marrow samples were collected from the posterior iliac crest under local anesthesia. Plasma cell purification was conducted using monoclonal mouse anti–human CD138 antibody through immunomagnetic bead selection, using the AutoMACS automated separation system (Miltenyi-Biotec, Bergisch Gladbach, Germany). Samples with a post-sort purity exceeding 80% were chosen via flow cytometry. Subsequently, RNA extraction was carried out, and gene expression profiling was performed (refer to the Appendix A). The patients’ GEP70, GEP80, proliferation index, and Sky92 scores were calculated. Chromosome translocations (4;14 and 14;16) were determined based on the spikes observed in the FGFR3 and c-MAF expression levels, respectively.

### 2.3. Fluorescence In Situ Hybridization

To remove erythrocytes from bone marrow aspirates, a Ficoll–Hypaque gradient-centrifugation separation technique was performed. To identify TP53 deletions associated with 17p del, a SpectrumRed-labeled DNA probe (LSI p53; Vysis, Downers Grove, IL, USA) and a SpectrumGreen-labeled probe (CEP17, Vysis), targeting the α-satellite DNA centromere of chromosome 17, were utilized. To detect 1q gain, bacterial artificial chromosomes (BACs) RP11-307C12 located at 1q21 and RP11-32D17 located at 1q31 were obtained from BAC/PAC Resources (Oakland, CA, USA). The Appendix A provides detailed information on these procedures. Cytogenetic subset information is supplied in Figure 1.

### 2.4. Statistical Analysis

For the remaining 29 variables, the random forest imputation method was used for data imputation [22]. In the univariate Cox regression, continuous variables were classified based on clinical standards, and ALB and B2M were transformed into ISS stage. Variables with a *p*-value < 0.05 were transformed into binary variables and included in the multivariate Cox regression. We finally selected seven understandable and well-distributed variables (whose abnormality represent >20% poor prognosis), and a weighted score was calculated based on their hazard ratio. The primary 4-stage system was defined via K-adaptive partitioning [23] and then adjusted slightly for the even distribution of each stage. Harrell’s c-index [24] was used for model comparisons and evaluation. The clear separation of the four stages was confirmed in validation cohorts, and a detailed description process is provided in the Appendix A.

### 2.5. Data-Sharing Statement

After the publication, data collected for this analysis and related documents will be made available to others. Data sharing requests need to be written and addressed to the attention of the corresponding author, Dr. Fenghuang Zhan, at the following e-mail address: fzhan@uams.edu.

## 3. Results

### 3.1. Subsection

#### 3.1.1. Patient Information

After filtering with PFS and OS information, a total of 5259 MM patients receiving ASCT were included in this analysis. The median follow-up period was 57 months. Details of variable statistics are presented in Appendix A. A random forest multiple imputation method was used to impute missing data. Baseline information of original data and data after imputation are shown in Appendix A. The patients were randomly divided into a training set (70%) and a validation set (30%). Table 1 summarizes the baseline characteristics of the patients in both sets.

#### 3.1.2. Stage System Development

We started from the training set and transformed continuous variables into classified variables based on clinical standards (see Appendix A). Univariate Cox regression was performed in the training set (see Appendix A). Variables with *p* < 0.05 were transformed into binary and were included in the multivariate Cox regression analysis (Appendix A). Time from MM diagnosis to ASCT, platelet count, B2M, ferritin, transferrin, LDH, calcium, age, and gender were found to be strong risk factors for both PFS and OS.

We selected independent prognostic factors to develop the stage system. We started with B2M variable (B2M > 5.5 represents ISS stage III) and added other independent prognostic factors into the Cox model. Platelets, calcium, and isotypes were excluded from the final model since they only represented poor prognosis in fewer than 20% of patients (Appendix A). As more variables were added, the c-index of the model increased and eventually plateaued (Figure 2A,B, Appendix A). Ultimately, seven variables were selected for the model: ISS, diagnosis–transplant period, ferritin, LDH, transferrin, age, and gender. We fitted the Cox model using only these seven variables and normalized the hazard ratio to ISS stage III compared to ISS stage I and II, choosing the closest integer or integer plus 0.5 as the score for each variable (Figure 2C). Then, risk scores were calculated for all patients, and k-adaptive partitioning was used to define stage borders, just like the R-ISS model [20]. We started from the four-stage system generated by k-adaptive partitioning, adjusted the border for an even distribution (Appendix A), and finally obtained the autologous transplant myeloma four-stage system (ATM4S) (Figure 2D–F). Patients in stage II, III, and IV have a higher hazard ratio than those in stage I (for PFS, stage II: HR [hazard ratio]: 1.452, 95%CI [confidence interval]: [1.296, 1.627], *p* < 0.0001; stage III: HR: 1.975, 95%CI: [1.771, 2.202], *p* < 0.0001; stage IV: HR: 3.476, 95%CI: [3.070, 3.935], *p* < 0.0001. For OS, stage II: HR: 1.620, 95%CI: [1.433, 1.832], *p* < 0.0001; stage III: HR: 2.366, 95%CI: [2.104, 2.660], *p* < 0.0001; stage IV: HR: 4.650, 95%CI: [4.077, 5.304], *p* < 0.0001). To further assess the proportionality hazard assumption, we conducted log-negative log plots on the training dataset. The results indicate that each stage exhibits parallel curves, both among themselves and relative to one another, thereby affirming the fulfillment of the proportionality hazard assumption within the stage system (Figure 2G,H).

#### 3.1.3. Model Generalization Performance in UAMS Validation Set

Then, we calculated scores and generate the stages of 1576 ASCT MM patients. Kaplan–Meier curves show clear curve separation for the four stages (Figure 3A, Appendix A). Just like in the training set, patients in stage II, III, and IV have a higher hazard ratio than those in stage I (for PFS, stage II: HR: 1.506, 95%CI: [1.257, 1.805], *p* < 0.0001; stage III: HR: 2.054, 95%CI: [1.728, 2.440], *p* < 0.0001; stage IV: HR: 3.647, 95%CI: [2.976, 4.470], *p* < 0.0001. For OS, stage II: HR: 1.767, 95%CI: [1.456, 2.143], *p* < 0.0001; stage III: HR: 2.466, 95%CI: [2.049, 2.968], *p* < 0.0001; stage IV: HR: 4.685, 95%CI: [3.777, 5.810], *p* < 0.0001). Log-negative log plots were also performed on the training dataset for conducting the proportion hazard test (Figure 3B, Appendix A). These plots generally satisfy the proportionality hazard assumption.

To evaluate the consistency of the four-stage system in the training and validation sets, we compared the survival curves in the training and validation sets in each stage (Figure 3C–F, Appendix A). In all stages, including both PFS and OS, no statistically significant differences were observed between the training and validation sets. This suggests a consistent predictive effect across both sets.

Additionally. we evaluated ATM4S in different clinical trial subsets [25,26,27,28,29]. Among the 5259 patients in the UAMS cohort, six clinical trials were found (Total Therapy 1–6). Detailed information of these clinical trials is presented in Appendix A. These clinical trials were divided into three groups: Total Therapy 1, Total Therapy 2–4, and Total Therapy 5–6. Total Therapy 1 includes all patients receiving ASCT. Total Therapy 2–4 covers ASCT with and without IMiD (immunomodulatory drug) and ASCT with IMiD and PI. Total Therapy 5–6 includes the stratification of patients based on GEP70-defined low- and high-risk MM, along with ASCT with ImiD and PI, with further modifications for GEP70 high-risk MM. ATM4S can separate most of the stages in these clinical trial subsets (Appendix A). In Total Therapy 1, stage IV has few patients and the other stages had good separation. In GEP70 high-risk MM, patients in stage II and stage III had similar prognoses in overall survival. To further explore the added predictive value of cytogenetic variables, subsets with FISH-defined chromosome abnormalities and GEP were extracted. We conducted an assessment of the prognostic significance of various chromosomal aberrations, including deletion (17p del) [30], gain (1q gain) [31], and translocation (4;14, 14;16) [32], as well as gene expression profile (GEP) scores within specific cytogenetic subsets (Appendix A). The incorporation of these genetic signatures demonstrated an enhancement in the c-index, indicating improved prognostic accuracy.

#### 3.1.4. ATM4S Performance at Iowa Medical Center

ATM4S was further validated in a 514 ASCT MM cohort from the University of Iowa Medical Center (Figure 4A,B). The patients in the four stages exhibited the anticipated differentiation in prognosis. Patients in stage II, III, and IV have a higher hazard ratio than tose stage I (for PFS, stage II: HR: 1.381, 95%CI: [0.9515, 2.005], *p* = 0.0894; stage III: HR: 1.662, 95%CI: [1.165, 2.371], *p* = 0.0051; stage IV: HR: 2.629, 95%CI: [1.711, 4.040], *p* = 0.0001. For OS, stage II: HR: 1.548, 95%CI: [0.969, 2.472], *p* = 0.0675; stage III: HR: 1.592, 95%CI: [1.075, 2.663], *p* = 0.0023; stage IV: HR: 2.805, 95%CI: [1.644, 4.786], *p* = 0.0002).

## 4. Discussion

This study includes easy and fast clinical variables, such as diagnosis–transplant period, age, gender, ISS stage, LDH, ferritin, and transferrin, to generate the ATM4S. Unlike all previous myeloma prognostic models, we set the transplant date as the starting date used in the model development. The ATM4S serves as a prognostic model to help clinicians determine the risk stratification of ASCT for MM patients. Most of the patients used in this study were from UAMS. The time span for MM diagnosis ranges from 1989 to 2022, which is quite long. So, we tested the four-stage system in three-time-period subsets (the oldest 1/3, median 1/3, and latest 1/3 ASCT MM patients’ cohort), and it works in all three diagnosis periods (Figure 5). Fewer than 2% of the 5259 patients used in this study were diagnosed after 2020, ensuring that the results are not significantly biased by limited follow-up. We also found that including genetic variables improved the performance of ATM4S.

The application of autologous stem cell transplantation (ASCT) in multiple myeloma (MM) dates to 1989 and involves induction chemotherapy, stem cell collection, high-dose chemotherapy, stem cell reinfusion, bone marrow reconstruction, and maintenance. There are some guidelines with which to evaluate whether patients are suitable for ASCT therapy, such as age, performance status, and comorbidities (especially kidney dysfunction), which are used for short-term risk events. For long-term benefits, there have been some risk factor studies, including lymphocytes after ASCT [33], expression of PARP1 and POLD2 [34], very good partial response (VGPR) [35], minimal residual disease [36,37], and PET/CT scan near day 100 post-ASCT [38]. These indices, whether before or after ASCT, provide information about ASCT patients’ long-term benefits. Some general indices for myeloma can also predict the progression-free survival and overall survival of MM patients receiving ASCT, but it is difficult to evaluate the benefits of ASCT since the start date in these studies is the first treatment date. Here, we aimed to predict the long-term benefits of ASCT by using all standard variables taken prior to ASCT to classify patients.

Upon analyzing the 29 variables, we identified three prognostic factors for autologous stem cell transplantation (ASCT) in multiple myeloma (MM) that are not currently incorporated into existing MM prognostic models. These factors include the time interval from MM diagnosis to transplant, ferritin levels, and transferrin levels. Patients undergoing ASCT one year or more after MM diagnosis generally exhibited a poorer prognosis. Within this subgroup, some underwent prolonged induction therapy, while others received salvage transplants. Elevated ferritin levels were also established as a significant adverse prognostic factor. Elevated ferritin levels may be associated with inflammation, increased iron burden, liver disease, tumor burden, and other factors [39]. Additionally, we identified low transferrin levels, indicative of a poorer MM prognosis. Transferrin is responsible for iron transportation, and low transferrin levels are commonly associated with malnutrition, liver disease, inflammation, thyroid dysfunction, and tumors. Further exploration is warranted to elucidate the origins of serum ferritin and transferrin, as well as the underlying biological processes contributing to elevated ferritin and decreased transferrin in MM patients [40].

In the ATM4S system, 85% of MM is classified into stages I, II, and III, with a median progression-free survival of 65, 51, and 41 months, and a median overall survival of 90, 69, and 55 months, respectively. The remaining 15% of patients in stage IV have a relatively poor prognosis, with a median progression-free survival of 20 months and a median overall survival of 28 months. It remains uncertain as to whether patients at stage IV could experience enhanced survival benefits by embracing alternative therapeutic strategies such as CAR-T cell therapy and other immunotherapies.

Starting with the widely used ISS stage, we used a stepwise incorporation of variables to create the ATM4S staging system. Gender, age, and diagnosis–transplant period are independent prognostic factors in the ATM4S that can be determined without any testing. In addition, three independent serum variables—LDH, ferritin, and transferrin—were contained in ATM4S. Interestingly, high serum ferritin and iron and low transferrin are high-risk factors. The sources of ferritin and iron and the mechanisms of transferrin loss is currently unknown and will be the focus of future research [41,42].

There are also some limitations in our study. For instance, we have not explored the significance of other important factors, such as disease status at the time of autologous stem cell transplantation (ASCT), the presence of extramedullary disease at initial diagnosis and during ASCT, plasma cell morphology, the severity of bone disease, the type of initial induction therapy, and the type of ASCT conditioning regimen. For clinical indexes, serum uric acid level and other liver function indexes should also be evaluated. Additionally, in our current dataset, data regarding “treatment-related mortality events” are not included. The causes of patient mortality will be systematically analyzed in our subsequent research. Furthermore, comprehensive data on the subsequent treatment regimens administered to patients experiencing post-transplantation relapse should be collected. Future research endeavors could delve into these areas to provide a more comprehensive understanding of predictive factors for ASCT outcomes in multiple myeloma patients.

## 5. Conclusions

In conclusion, our study introduces the autologous stem cell transplantation myeloma staging system (ATM4S), incorporating novel prognostic factors for patients with multiple myeloma (MM) undergoing ASCT. High serum ferritin and iron, along with low transferrin, emerge as significant risk factors. Additionally, the inclusion of genetic signatures enhances prognostic accuracy. The ATM4S offers valuable insights for risk stratification, guiding clinicians in personalized treatment decisions. This comprehensive staging system represents a significant advancement in predicting long-term benefits and opens avenues for further research in MM management.

## Figures and Tables

**Figure 1 cancers-16-01116-f001:**
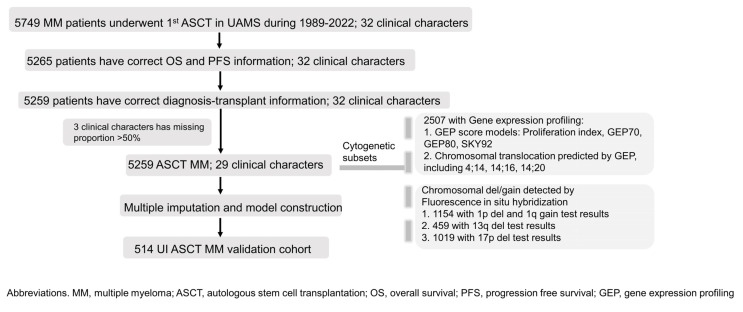
Flow chart of patient selection and cytogenetic subsets’ information. We initiated the study with a cohort of 5749 ASCT patients at UAMS. After excluding individuals lacking PFS and OS data, 5259 patients remained. Initially, 32 clinical variables were selected, with 3 being omitted due to high missing data rates. The final cohort consisted of 5259 patients with 29 clinical variables. Missing data were addressed via multiple imputation, enabling the development of a 4-stage prognostic system. External validation was performed using a separate 514-patient cohort from the University of Iowa. In the cohort of 5259 patients, 2507 had available gene expression profiling data. Among these, 1154 patients were tested for 1p deletion and 1q gain, 459 for 13q deletion, and 1019 for 17p deletion.

**Figure 2 cancers-16-01116-f002:**
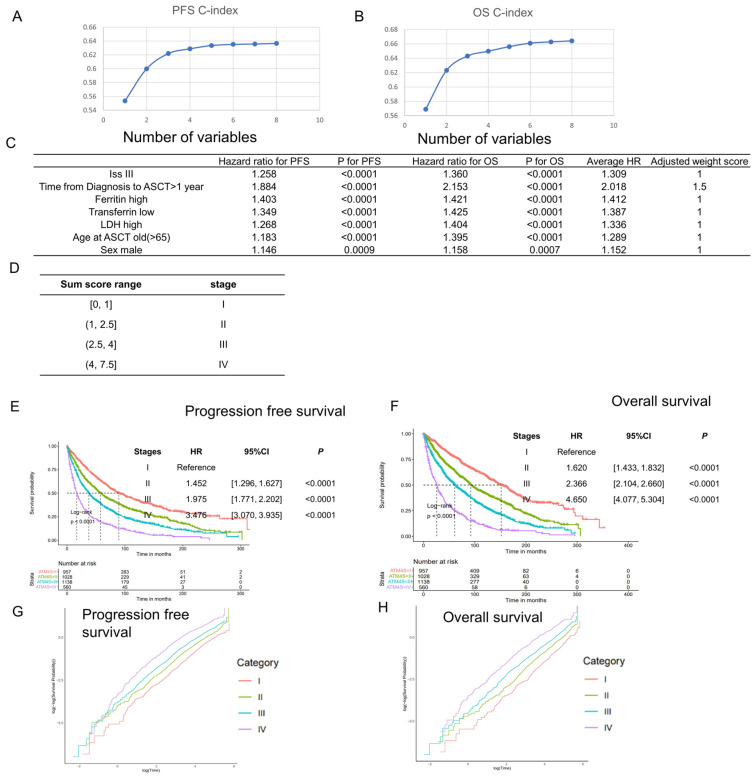
Staging system development in training set. (**A**,**B**) Harrel’s c-index of Cox models using PFS and OS as additional prognostic factors increased. Variables were sequentially added in accordance with the order depicted in Figure 2C. (**C**) The hazard ratios of the primary prognostic factors were utilized for model development and standardization. Each variable received a risk score determined by its hazard ratio. (**D**) Score border of each ATM4S risk stage. (**E**,**F**) Progression-free and overall survival curves of four stages in ATM4S. Higher stages represent poor prognosis. (**G**,**H**) Log-negative log plot of different stages. The survival curves of each group exhibit a parallel pattern, suggesting that they meet the proportional hazard assumption.

**Figure 3 cancers-16-01116-f003:**
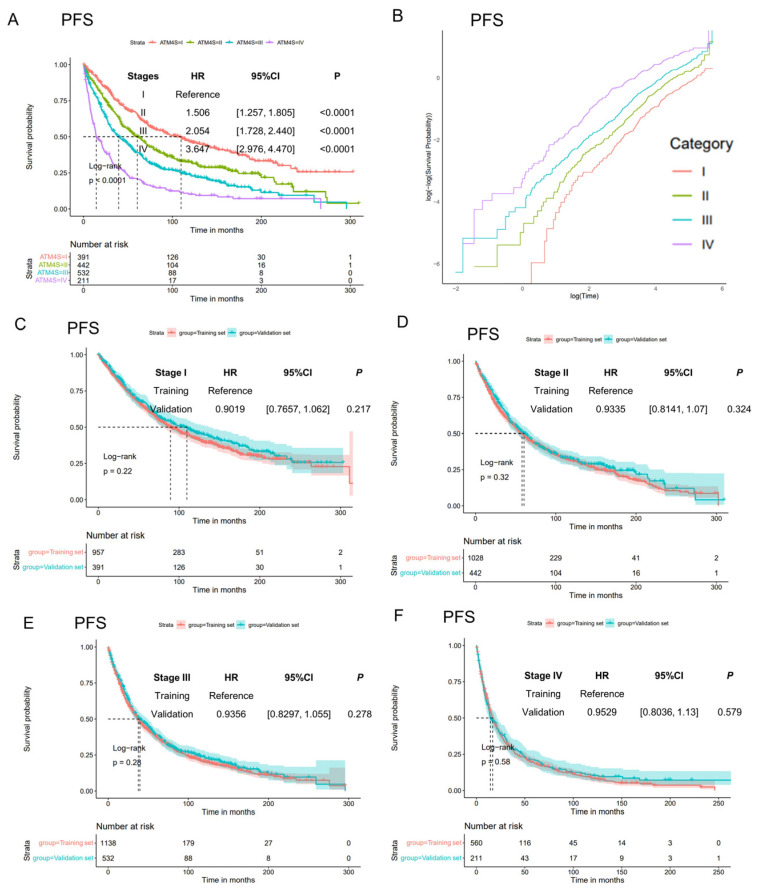
Assessment of model generalization performance. (**A**) Progression-free survival curves of ATM4S in UAMS validation cohort. The four-stage system demonstrates an anticipated distinction in prognosis. (**B**) Log-negative log plot of different stages. The survival curves of each group exhibit a parallel pattern, suggesting that they meet the proportional hazard assumption. (**C**–**F**) Comparison of progression-free curves for each stage in both the training and validation datasets. No statistically significant differences were observed within each stage, suggesting a consistent effect of the stage system in both datasets. The results of overall survival curves were shown in Appendix A.

**Figure 4 cancers-16-01116-f004:**
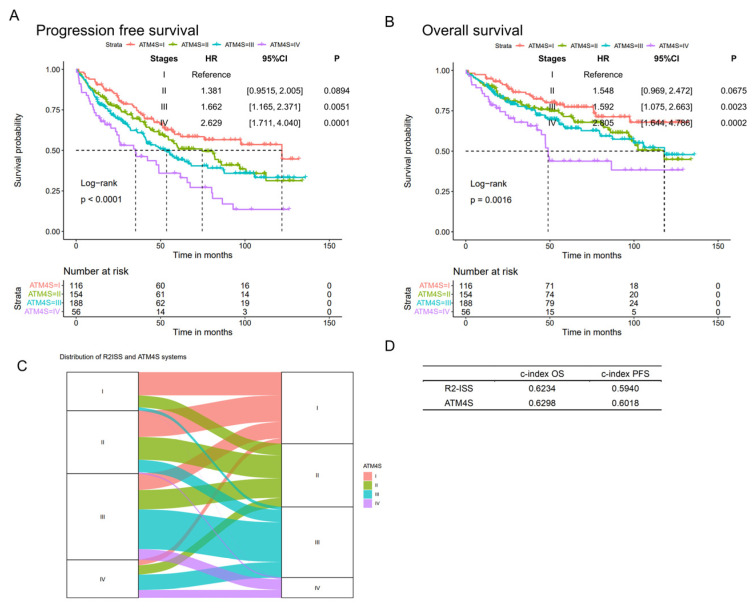
Validation in University of Iowa Cohort. (**A**,**B**) Progression-free and overall survival curves of ATM4S in University of Iowa validation cohort. The four-stage system demonstrates an anticipated distinction in prognosis. (**C**). An impact plot illustrating the allocation of ATM4S and R2ISS staging systems within a subgroup of 860 individuals in UAMS. (**D**). Harrel’s c-indexes of R2ISS and ATM4S were calculated in the 860-subgroup. Both staging systems exhibit comparable c-index values.

**Figure 5 cancers-16-01116-f005:**
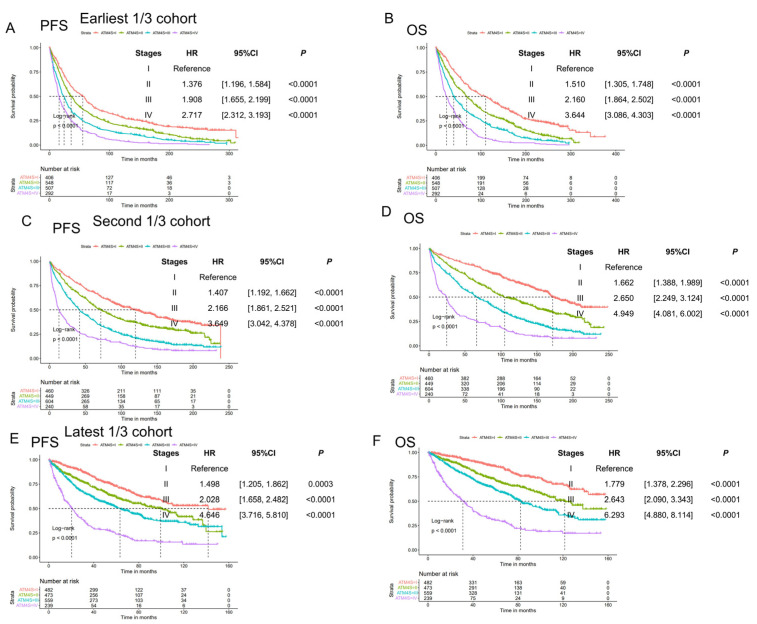
Validation in three different MM diagnosis periods: (**A**,**B**) progression-free and overall survival curves of ATM4S subgroups in the earliest 1/3 cohort; (**C**,**D**) progression-free and overall survival curves of ATM4S subgroups in the second 1/3 cohort; (**E**,**F**) progression-free and overall survival curves of ATM4S subgroups in the latest 1/3 cohort. In all periods, ATM4S can separate survival curves of ASCT MM.

**Table 1 cancers-16-01116-t001:** Baseline information of training and validation set.

Variables	Training (*n* = 3683)	Validation (*n* = 1576)
Sex, *n* (%)		
Female	1448 (39)	633 (40)
Male	2235 (61)	943 (60)
Age at transplant, Median (Q1, Q3), yr	59.55 (51.66, 66.27)	59 (51.67, 66.22)
Race, *n* (%)		
Asian	13 (0)	10 (1)
African	437 (12)	185 (12)
Native American	9 (0)	11 (1)
Pacific islander	2 (0)	1 (0)
White/Caucasian	3235 (88)	1369 (87)
Isotype, *n* (%)		
Biclonal disease	6 (0)	5 (0)
Free light chain	671 (18)	265 (17)
IgA	723 (20)	328 (21)
IgD	51 (1)	14 (1)
IgG	2025 (55)	880 (56)
IgM	13 (0)	7 (0)
Non-secretory	194 (5)	77 (5)
Light, *n* (%)		
Kappa	2246 (61)	942 (60)
Kappa + Lambda	4 (0)	2 (0)
Lambda	1256 (34)	567 (36)
None	177 (5)	65 (4)
Transferrin, Median (Q1, Q3), g/L	215 (177, 249)	213.5 (179, 245.25)
Ferritin, Median (Q1, Q3), μg/L	233.4 (100.4, 533.2)	254.8 (100.97, 556)
Iron, Median (Q1, Q3), μg/dL	73 (52, 98)	71 (51, 96)
Plasma cell percentage (bone marrow aspiration), Median (Q1, Q3), %	25 (7.5, 50)	27 (7.5, 50)
Plasma cell percentage (bone marrow biopsy), Median (Q1, Q3), %	30 (7.5, 60)	30 (7.5, 60)
Albumin, Median (Q1, Q3), g/dL	3.9 (3.5, 4.3)	3.9 (3.4, 4.3)
B2M, Median (Q1, Q3), mg/L	3.1 (2.1, 5.3)	3.2 (2.2, 5.23)
LDH, Median (Q1, Q3), U/L	158 (128, 200)	155.5 (127, 194.25)
Creatinine, Median (Q1, Q3), mg/dL	1 (0.8, 1.3)	1 (0.8, 1.3)
CRP, Median (Q1, Q3), mg/dL	2.42 (0.45, 5.5)	2.9 (0.43, 5.4)
Hb, Median (Q1, Q3), g/dL	11.4 (9.8, 12.8)	11.3 (9.8, 12.72)
Platelets, Median (Q1, Q3), 10^3^/μL	223 (170, 278)	219 (169, 278)
Monocytes, Median (Q1, Q3), %	8.9 (6.85, 11.7)	8.9 (6.88, 11.5)
Lymphocytes, Median (Q1, Q3), %	26.4 (18.2, 35.15)	26.3 (18.9, 35.2)
Serum M protein, Median (Q1, Q3), g/dL	1.4 (0.09, 3.5)	1.6 (0.2, 3.7)
Urine M protein, Median (Q1, Q3), g/L	0 (0, 504)	0 (0, 485.25)
Ca, Median (Q1, Q3), mg/dL	9.2 (8.8, 9.7)	9.2 (8.8, 9.7)
BMI, Median (Q1, Q3), kg/m^2^	27.94 (24.81, 31.62)	27.94 (24.8, 31.48)
Glucose Serum, Median (Q1, Q3), mmol/L	101 (90, 119)	100 (89, 119)
Cholesterol, Median (Q1, Q3), mg/dL	172 (138, 207)	172 (138, 207)
Triglycerides, Median (Q1, Q3), mg/dL	141 (93, 208)	141 (94, 215)
HDL, Median (Q1, Q3), mg/dL	42 (33, 53)	41 (33, 52.25)
Time from MM diagnosis to ASCT, Median (Q1, Q3), mth	6.53 (4.13, 11.97)	6.47 (4.19, 12.08)
OS time, Median (Q1, Q3), mth	56.1 (22.78, 112.52)	59.2 (25.37, 112.55)
OS, *n* (%)		
0	1326 (36)	594 (38)
1	2357 (64)	982 (62)
PFS time, Median (Q1, Q3), mth	38.77 (14.77, 84.08)	40.95 (16.72, 86.43)
PFS, *n* (%)		
0	1113 (30)	517 (33)
1	2570 (70)	1059 (67)

Abbreviations: Q1, first quantile; Q3, third quantile; B2M, beta-2 microglobulin; LDH, lactate dehydrogenase; CRP, C-reactive protein; Hb, hemoglobin; BMI, body mass index; HDL, high-density lipoprotein; OS, overall survival; PFS, progression-free survival.

## Data Availability

After the publication, data collected for this analysis and related documents will be made available to others. Data requests need to be written and addressed to the attention of the corresponding author, Fenghuang Zhan, at the following e-mail address: fzhan@uams.edu.

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
