# Peer review of "A Risk Stratification System in Myeloma Patients with Autologous Stem Cell Transplantation"

_cancers, 2024, doi:10.3390/cancers16061116_

Round 1
Reviewer 1 Report
Comments and Suggestions for Authors
This is an interesting and very well-designed retrospective study examining predictive factors for progression-free survival in a cohort with an impressive number of multiple myeloma patients undergoing autologous stem cell transplantation, which has been validated in a cohort of similar patients from Iowa. The model includes the variable 'time from diagnosis to transplantation' >1 year, a possible indicator of low response disease, as a predictor with the highest adjusted weight score of 1.5, but the four categories are well separated and stage IV is defined by a sum score range of 4-7.5. Other variables are important in determining high-risk patients, among them the authors indicate ferritin, transferrin and LDH levels as predictive factors, easily assessable at diagnosis. Interestingly, male gender, ISS score of 3 and age over 65 years already identify stage III disease, and only one additional factor is sufficient to reach the very poor prognosis group of stage IV patients, with the highest HR of 3.476 corresponding to a median PFS of 20 months. The PFS survival curve of the Iowa cohort appears to show a slightly higher median PFS than these results in stage IV group. This could be related to a possible era effect or to the availability of new drugs in the Iowa cohort. Can you comment this?
Author Response
Thank you sincerely for acknowledging our study. We truly appreciate your thoughtful questions concerning treatment duration and patient outcomes. In response, we are committed to gathering more comprehensive patient treatment information to better understand the nuances affecting differences in patient prognosis. It's worth noting that our data collection for the Iowa cohort began after 2013, while that of the Arkansas cohort commenced as early as 1989. This temporal disparity suggests that advancements in therapy, integrated into transplantation protocols over time, may have contributed to the observed differences in progression-free survival within the Iowa cohort. Your interest in our research is invaluable, and we look forward to further discussion and exploration in this area.
Reviewer 2 Report
Comments and Suggestions for Authors
In this manuscript the authors have performed a retrospective analysis of the prognostic factors, affecting the outcome of autologous SCT in patients with myeloma. The study is well designed and has used a validation cohort to confirm the accuracy and importance of its findings. In a period in which the value of ASCT in myeloma starts to be disputed, such an analysis might be useful to help the recognition of patients at higher transplantation risk, for whom an alternative treatment approach might be applied. The manuscript merits publication but needs to be improved at some points. Introduction is very short and poorly informative. In this section the authors need to report in more details similar studies previously published on prognostic factors, predicting ASCT outcomes among myeloma patients. Instead they only mention all previous experience in just 3 lines (57-59) and only provide 7 relevant references. They did not mention studies evaluating the clear need of transplantation vs no ASCT, in some of which risk factors for the transplantation outcomes have been recognized. They should also comment on the non recognition of serum LDH levels as a risk factor, although this has been reported by other groups. The significance of some other important factors, such as disease status at ASCT, presence of extramedullary disease at initial diagnosis and at ASCT, plasma cell morphology, severity of bone disease, type of initial induction treatment and type of ASCT conditioning regimen have not been investigated. Even when these factors may not be available in the whole patient population, they can still exist in a substantial proportion of the more than 5,000 patients, which has been analyzed. At least these factors should have been mentioned, and should had been compared with the findings obtained in the current study and finally they need to be discussed in the discussion part.
Finally the provided reference list needs to be enriched with other informative studies on prognostic factors of myeloma patients, even outside the setting of ASCT.
MINOR ISSUES
Lines 72-73: This sentence has already mentioned in the previous paragraph-last of the introduction and its repetition here offers nothing.
Lines 73-74: Significance of Serum uric acid levels and normal vs abnormal liver function tests should also had been analyzed.
Reviewer 3 Report
Comments and Suggestions for Authors
It is an extensive study of factors important for PFS and OS in Multiple myeloma. Factors were collected and studied at the time of Autologous transplantation. New factors appeared important (ferritin, transferrin and the time diagnosis-autologous transplant; thus, on this basis, a new original classification was rigorously constructed and extensively validated. To possibly improve this paper I have some suggestions:
1) Add the time point before autologous transplantation in which the factors were studied (at least ranges).
2) Specify the source of origin of marrow plasmacytosis (if aspirate or Core Biopsy or either of two)
3) Comparison of ISS staging and ATM4S and cross-tabulation is provided in Figure 4 (Iowa cohort), it is an important piece of information. Thus, makes this figure well understandable.
4) GEP is described in supplemental materials and its preliminary results are reported in the discussion section while no results are reported in result section. Consider moving part of this GEP paragraph from discussion to the Result Section.
5) avoid generic statements (such as "meticulously" divided; "rigorous" validation) in fact, the high rigour of your methodology is already evident.
6) Some log-rank test for groups looks to be "trend log-rank"; please specify.
7) HR resulted higher for OS than for PFS. Your system may be able to predict TRM ? or response to second-line treatment in patients relapsing after a first auto? You can consider to discuss this issue.
8) Specify in the inclusion criteria that all patients included were studied before their "first" autologous transplant.
9) Consider adding appropriate references when you describe Random Forest Imputation and K adaptive partitioning.
